# Calculated Whole Blood Viscosity and Albumin/Fibrinogen Ratio in Patients with a New Diagnosis of Multiple Myeloma: Relationships with Some Prognostic Predictors

**DOI:** 10.3390/biomedicines11030964

**Published:** 2023-03-21

**Authors:** Melania Carlisi, Rosalia Lo Presti, Salvatrice Mancuso, Sergio Siragusa, Gregorio Caimi

**Affiliations:** 1Department of Health Promotion, Mother and Child Care, Internal Medicine and Medical Specialties, University of Palermo, 90127 Palermo, Italy; 2Department of Psychology, Educational Science and Human Movement, University of Palermo, 90127 Palermo, Italy

**Keywords:** multiple myeloma, hemorheological pattern, calculated blood viscosity, albumin/fibrinogen ratio, prognostic factors

## Abstract

Background: In this single center study, we retrospectively evaluated the calculated hemorheological profile in patients with a new diagnosis of multiple myeloma, with the aim to evaluate possible relationships with some prognostic predictors, such as ISS, albumin levels, beta2-microglobulin, red cell distribution width, and bone marrow plasma cell infiltration. Methods: In a cohort of 190 patients, we examined the calculated blood viscosity using the de Simone formula, and the albumin/fibrinogen ratio as a surrogate of erythrocyte aggregation, and then we related these parameters to prognostic factors, using the Kruskal–Wallis and the Mann–Whitney tests, respectively. Results: From our analysis, it emerged that the evaluated hemorheological pattern differed in the three isotypes of multiple myeloma, and the whole blood viscosity was higher in IgA and IgG isotypes with respect to the light chain multiple myeloma (*p* < 0.001). Moreover, we observed that, as the ISS stage progressed, the albumin/fibrinogen ratio was reduced, and the same hemorheological trend was traced in subgroups with lower albumin levels, higher beta2-microglobulin and red cell distribution width RDW values, and in the presence of a greater bone marrow plasma cell infiltrate. Conclusions: Through the changes in blood viscosity in relation to different prognostic factors, this analysis might underline the role of the hemorheological pattern in multiple myeloma.

## 1. Introduction

Multiple myeloma (MM) is the second most common hematologic cancer, with a median age at diagnosis of 65 years. The risk of developing MM is higher in older age groups, whereas the diagnosis is more uncommon in patients under the age of 45 [1]. MM is a disease characterized by the presence of abnormal clonal plasma cells in the bone marrow, with potential uncontrolled growth causing destructive bone lesions, kidney injury, anemia, and hypercalcemia [2].

Despite improvements in treatment [3], MM remains an incurable disease and the affected patients may have poor quality of life due to disease-related symptoms and adverse events from therapies with cumulative toxicity. For this reason, one of the current directions of MM research is the identification of prognostic markers to stratify the patients into specific risk groups, with a significant impact on the accurate prognosis assessment and the selection of an appropriate therapeutic approach [4,5,6].

Many prognostic biomarkers have been identified over the years. These markers reflect host factors, tumor-related factors, tumor stage, disease burden, and the tumor response to treatment.

Among these prognostic factors, the staging systems play a key role. The International Staging System (ISS), which has surpassed the Durie–Salmon classification, is widely available, and it is based on two simple and routine laboratory tests: serum albumin and beta2-microglobulin (beta2-MG). It is robustly validated and applicable across geographical areas [7]. Considering the remarkable role of cytogenetic alterations, in 2015, the Revised International Staging System (R-ISS) was introduced. This new prognostic system includes, in addition to albumin and beta2-MG, too high-risk chromosomal abnormalities detected by interphase fluorescence in situ hybridization (FISH) (deletion (17p), translocation t(4;14) (p16;q32), or t(14;16) (q32;q23)) and serum lactate dehydrogenase (LDH) levels [8,9]. However, this revised staging system has a limitation represented by the fact that 62% of patients were classified into the intermediate-risk category, when instead the patients could belong to different risk levels of progression/death. Recently, a second revision of the R-ISS, named R2-ISS, has been proposed. In this revision, 1q gain/amplification are included in the risk calculation [10].

In addition to ISS and its revisions, an important prognostic role in MM patients is also played by albumin levels alone. In fact, besides being used as a factor when calculating ISS, the albumin level has been found to be an independent predictor of mortality [11,12]. The same consideration can also be made for beta2-MG; elevated serum beta2- MG levels represent a tumor marker and may indicate the tumor burden in hematologic malignancies, especially in MM [13]. Additionally, beta2-MG is an independent predictor of survival in MM and an independent predictor of progression in patients with asymptomatic MM [14].

Furthermore, important prognostic information is also derived from the numerical and morphological assessment of bone marrow plasma cells (BMPC) [15,16], and from the pre-treatment red cell distribution width (RDW) [17].

Finally, in the MM setting, the hemorheological pattern can play a significant role, especially in terms of whole blood viscosity (WBV) and erythrocytes deformability and aggregation [18,19,20,21].

Generally, the measurement of WBV is carried out ex vivo using different types of viscometers, such as rotational, capillary, and oscillatory ones. However, WBV can also be indirectly calculated using specific formulas, from some simple laboratory parameters (such as hematocrit, total plasma proteins, and fibrinogen), overcoming the technical difficulties and the costs of use and maintenance of common viscometers [22,23,24,25]. In the de Simone formula, the calculation of WBV is obtained from the hematocrit and total plasma proteins. However, it must be emphasized that this formula was validated with specific reference ranges for the values of laboratory parameters.

Although these reference ranges are not often respected in MM patients, the possibility of using the calculated whole blood viscosity (c-WBV) in plasma cell dyscrasias was provided by the results of our recent clinical study in which, when evaluating the directly measured and calculated blood viscosity data in monoclonal gammopathy of undetermined significance (MGUS) and MM patients, it emerged that the c-WBV and the surrogate marker of erythrocyte aggregation showed a parallel trend [26].

In addition, this formula does not consider the deformability and the aggregation of erythrocytes, but the latter parameter may be indirectly calculated using the albumin/fibrinogen ratio, as mentioned above [27,28].

Therefore, considering the possible role of the hemorheological profile in MM, we performed a single center retrospective analysis with the aim to evaluate, in a cohort of patients with a new diagnosis of MM (NDMM), eventual associations between the c-WBV and albumin/fibrinogen ratio with some recognized prognostic predictors. The analysis was performed in the entire group of patients and by dividing the sample into three subgroups based on the MM isotype: light chains MM, IgA, and IgG MM (LCMM, IgA MM, and IgG MM). The evaluation of the hemorheological pattern was carried out in a calculated way, using the de Simone formula for the c-WBV and the albumin/fibrinogen ratio as a surrogate of erythrocyte aggregation.

## 2. Materials and Methods

### 2.1. Population

This is a retrospective single center study performed on 190 patients (102 women and 88 men; average age 69 ± 10) with a new diagnosis of MM, evaluated at the Hematology Division of the “Paolo Giaccone” University Hospital in Palermo between 1 January 2017 and 30 September 2022. Specifically, the sample is comprised of 107 patients with IgG MM (71 with IgG κ and 36 IgG λ isotype), 56 patients with IgA isotype (28 with IgA κ and 28 IgA λ isotype), and 27 patients with LCMM (13 with expression of κ light chain and 14 with λ light chain). Patients over the age of 18 who had a set of investigations performed on serum, urine, and bone marrow samples, who were indicated at the diagnosis for the evaluation of the underlying disease, were included. 

### 2.2. Laboratory Tests

In this retrospective study, we evaluated the following parameters: hematocrit (Ht), obtained using an automated hematology analyzer; total plasma proteins, expressed in g/L and evaluated with the colorimetric method; fibrinogen, expressed in g/L and evaluated with the Clauss method; WBV at 208 s^–1^, calculated according to the de Simone formula ((0.12 × Ht) + 0.17(TP − 2.07)); albumin (g/L), evaluated using the colorimetric method; and, finally, the albumin (g/L)/fibrinogen (g/L) ratio.

### 2.3. Statistical Analysis

All of the statistical analyses were performed using GraphPAd Prism version 9.5. The data were expressed as medians and interquartile ranges. The one-way variance analysis, concerning the comparison between the different isotypes and between different stages of ISS, was performed using the Kruskal–Wallis test, integrated with the Dunn test. The median comparison was made using the Mann–Withney test. The correlation coefficient for Spearman ranks was used for the analysis of the different correlations. The null hypothesis was evaluated for values of *p* ≤ 0.1.

## 3. Results

The clinical data and baseline characteristics of the study population are summarized in Table 1. In the entire cohort of MM patients, we first evaluated the medians, interquartile ranges, and ranges of the hemorheological pattern (Table 2). Then, by comparing the whole blood viscosity and the remaining parameters in the MM isotypes, we observed that the c-WBV, total plasma proteins, albumin, and fibrinogen levels distinguished the three isotypes; particularly, the c-WBV and total plasma proteins were higher in the IgA and IgG MM with respect to LCMM, while the albumin level was reduced in the IgA and IgG isotypes in comparison with the LCMM (Table 3).

Dividing the entire cohort of MM patients based on ISS, the hematocrit value, albumin level, and albumin/fibrinogen ratio decreased from stage I to stage III, while the total plasma proteins increased; no changes in c-WBV and plasma fibrinogen were observed (Table 4).

In the whole cohort and in the three MM isotypes, we then calculated the medians of some prognostic predictors: albumin (all MM 36.39 g/L, LCMM 39.90 g/L, IgA 33.85 g/L and IgG 37.00 g/L), beta2-MG (all MM 4.70 mg/L, LCMM 4.40 mg/L, IgA 4.490 mg/L and IgG 4.70 mg/L), RDW (all MM 15.1%, LCMM 14.4%, IgA 16% and IgG 14.8%), and BMPC (all MM 40%, LCMM 40%, IgA 60% and IgG 30%) (Appendix A). Therefore, in the entire group of patients and in the different isotypes, the behavior of c-WBV and the other hemorheological parameters was evaluated in relation to these medians.

Dividing the whole study population according to the albumin level, in the subgroup with lower albumin values, we observed a reduction in hematocrit and albumin/fibrinogen ratio, and an increase in the total plasma proteins (Table 5). The same subdivision was carried out in patients with LCMM and, always in the group with lower levels of albumin, we observed a reduction in hematocrit, total plasma proteins, and c-WBV (Appendix A). Evaluating the IgA isotype, there was observed a reduction in the hematocrit and albumin/fibrinogen ratio, and an increase in total plasma proteins and c-WBV (Appendix A); however, for the IgG isotype, lower levels of albumin were associated with a decrease in hematocrit values and in the albumin/fibrinogen ratio (Appendix A).

The distribution of the whole cohort of patients according to the beta2-MG levels, in the subgroup that exceeded the median value, showed a reduction in the hematocrit, albumin, and albumin/fibrinogen ratio, and an increase in the total plasma proteins (Table 6). The same evaluation performed in LCMM showed only a reduction in hematocrit in the subgroup with higher levels of this predictor (Appendix A). No differences were observed in the IgA isotype (Appendix A), while in IgG MM, a reduction in hematocrit, albumin, and albumin/fibrinogen, and an increase in total plasma proteins were associated with higher levels of beta2-MG (Appendix A).

Still subdividing the entire group of patients according to the RDW, in the subgroup the exceeded the median, a decrease in hematocrit, albumin, and the albumin/fibrinogen ratio and an increase in total plasma proteins were observed (Table 7). In LCMM, the identical approach showed a reduction in hematocrit, c-WBV, and albumin levels (Appendix A). In the IgA isotype, values exceeding the median of RDW were associated with a reduction in hematocrit and albumin levels (Appendix A), and, in the IgG MM subgroup, with a decrease in hematocrit, albumin, and albumin/fibrinogen ratio (Appendix A).

Finally, making a subdivision of the entire cohort of MM patients according to the BMPC percentage, in the subgroup that exceeded the median values, we observed a reduction in hematocrit and albumin levels (Table 8). The same analysis performed in the LCMM group revealed a decrease in c-WBV in the presence of BMPC values beyond the median (Appendix A). In IgA MM, only a decrease in albumin levels was observed (Appendix A), while in the IgG isotype, a reduction in hematocrit and albumin levels and an increase in total plasma proteins and c-WBV was observed (Appendix A).

## 4. Discussion

In relation to the amount of data examined and considering that this retrospective single center study covered not only the whole group of MM patients but also the different MM isotypes, we mainly focused c-WBV (and the parameters that determine it) and the albumin/fibrinogen ratio (indirect indicator of erythrocyte aggregation) in the entire cohort of patients.

Indeed, it is necessary to consider that the different MM isotypes (LCMM, IgA, and IgG) are present in significantly different percentages, with the LCMM representing only 14% of the entire sample. Regarding c-WBV, the three isotypes are different, with values of c-WBV being higher in the IgA and IgG isotypes with respect to LCMM.

The increase in c-WBV, observed in IgA and IgG MM, is mainly due to the higher levels of total plasma proteins present in these isotypes, with no changes in the hematocrit. In our analysis, we found that in the entire cohort of patients, as well as in IgA and IgG isotypes, c-WBV is only related to total protein levels (data not shown), and this explains the behavior of this hemorheological parameter. However, in LCMM, c-WBV is dependent not only on the protein levels, but also on the hematocrit values (data not shown).

Moreover, variations in the albumin and fibrinogen levels are evident among the different isotypes. Albumin is a pivotal prognosis predictor in MM, and its reduced levels are associated with an early mortality (less than 12 months) [29]. Albumin has sharply different values in the three MM isotypes, and, in fact, its levels in IgA and IgG isotypes are reduced compared with those observed in LCMM. The plasma fibrinogen level has the same behavior, and it is significantly higher in LCMM. Instead, among the three different isotypes there is no appreciable variation in the albumin/fibrinogen ratio, which shows a particular trend in relation to the prognostic predictors mentioned above.

The subdivision of the whole cohort of MM patients according to ISS shows some interesting findings. Proceeding from stage I to stage III, the decrease in hematocrit is evident, associated with a slight increase in total plasma proteins, with no changes in c-WBV. The albumin/fibrinogen ratio, which is significantly reduced when progressing from stage I to III stage, behaves similarly to the albumin levels. Therefore, from a hemorheological point of view, the above-described shows that, with the progress of the ISS, MM patients become more anemic and at the same time register a theoretical tendency to erythrocyte aggregation. The increase in erythrocyte aggregation, evaluated with different methods, turns out to be a constant in the hemorheological pattern of patients with plasma cell dyscrasias [30,31].

Red cell aggregation has a key role, especially in areas of circulation where low sliding gradients prevail, such as the venous system [32]. This hemorheological determinant depends on the degree of interrelation between plasma proteins and erythrocytes, and it is a function of the concentration, size, and density of the plasma proteins, electrical charge surface expressed as the potential zeta, the plasma dielectric coefficient, and the membrane properties. Erythrocyte aggregation is a reversible process that affects blood viscosity at low shear rates and it certainly does not play any role in blood viscosity at high shear rates. At the same time, the monoclonal immunoglobulins at a non-physiological concentration can significantly interfere with the behavior of this determinant.

In the subdivision performed based on the albumin levels, the patients with values below the median showed a reduction in hematocrit, an increase in total plasma proteins, and a rise, although not significant, in c-WBV; a decrease in the albumin/fibrinogen ratio was also markedly present. In MM patients, the reduction in albumin levels does not seem to depend on age and/or gender, nor does it appear to be affected by liver and kidney function, by the presence of bone osteolytic lesions, Bence−Jones proteinuria, hypercalcemia, and body weight [33]. Actually, the reduction in albumin levels is due to the fact that interleukin-6 (IL-6), but above all the altered cytokine network present in MM, reduces the hepatic synthesis of albumin, approximately equal to 200 mg per kilogram of body weight per day [34,35,36]. Even if in most recent nomograms related to the prognostic stratification of these patients’ cytokines are not taken into account [37,38], other authors have included the cytokines in prognostic nomograms, in particular, for newly diagnosed MM patients [39].

The division of the entire cohort of MM patients according to beta2-MG, which in the condition of normal renal function reflects the whole mass of MM cells, shows a reduction in the hematocrit, albumin levels, and albumin/fibrinogen ratio, with an increase in total plasma proteins and fibrinogen in the subgroup exceeding the median value. Considering that some nomograms related to prognostic stratification simultaneously consider the levels of serum albumin and beta2-MG, we examined the correlation between these two parameters (Spearman test) in the entire cohort of patients, observing a negative correlation between the above parameters (r = −0.37 *p* < 0.001).

The results obtained with the subdivision of the whole cohort of patients according to the median of the RDW were similar. In fact, in the subgroup with higher RDW values, we observed a decrease in hematocrit, in albumin levels, and the albumin/fibrinogen ratio, associated with an increase in the total plasma proteins. Many papers in the literature underline the prognostic role of RDW in MM [17,40,41]. Indicating the heterogeneity of the volume of erythrocytes, RDW is a simple and immediately inflammatory marker and it reflects the increase of some cytokines, such as IL-6 and tumor necrosis factor-alpha (TNF-alpha), but also of hepcidin in the blood [42,43]. It is possible that the increase in this predictor may affect the erythrocyte deformability of MM patients, also considering that in some of our previous research [18,19,20,21], this hemorheological determinant was reduced compared with the control group. In this regard, in a cohort of 298 normal adults, some authors found a negative correlation between RDW and erythrocyte deformability, and only when the RDW value exceeded 14% was the reduction of this hemorheological determinant highlighted [44]. In addition to normal subjects, this correlation has been described in some hematological neoplasms [45].

The last subdivision, according to the percentage of BMPC, in the subgroup that overtook the median value, showed a significant reduction in the hematocrit and albumin levels. BMPC is a prognostic predictor in MM patients, and it is important to evaluate the cut-off in the BMPC percentage. In fact, a better prognosis was observed in patients with BMPC less than 50%, and this percentage was confirmed in several research works [46]. In more recent years, the issue of BMPC has been reconsidered by other authors, such as Qian in 2017 and Al Saleh in 2020 [47,48]. The latter hypothesized that a percentage of 60% predicted both disease-free survival and overall survival in MM patients. In relation to these recent data, we divided the entire study population, considering 60% BMPC as a limit, and we observed that in the 70 patients (37.4%) with values equal to or greater than the above percentage, in addition to the reduction of albumin and hematocrit, an increase in the total plasma proteins was also present (data not shown).

## 5. Conclusions

In conclusion, in this retrospective single center study, it is evident that c-WBV, with the same hematocrit values, was higher in the IgA and IgG isotypes with respect to the LCMM. It is interesting what happens in relation to the subdivision by ISS; in fact, as the stage progressed, we observed a reduction in albumin, an increase in the values of the total plasma proteins, and a significant decrease in the albumin/fibrinogen ratio, but, above all, a worsening of anemia. The same results were evident when performing the analysis with other prognostic predictors.

With the evaluation of c-WBV using the de Simone formula, it was not possible to obtain direct information on erythrocyte aggregation and deformability and plasma viscosity. However, using the albumin/fibrinogen ratio, information on erythrocyte aggregation can be reliable, and the reduction in this ratio was evident when patients were stratified on the basis of evaluated prognostic predictors. RDW was increased in MM patients and this marker could be related to erythrocyte deformability.

In our experience, erythrocyte deformability was reduced in MM patients and this datum mainly depends on the membrane dynamic properties and its lipid composition. Considering the above, it cannot be excluded the reduced red cell deformability may also be dependent on the increase in RDW. However, it must be considered that the albumin/fibrinogen ratio and RDW are strongly influenced by the altered cytokine network described in MM. In fact, the increase in IL-6 and TNF-alpha, also through the role played by hepcidin [49,50], interferes with erythropoiesis and thus with the RDW value, while the same cytokine network, inhibiting the hepatic synthesis of albumin, involves a decrease in the albumin/fibrinogen ratio.

As a limit, in our study, no information was available on the plasma viscosity (PV). However, in relation to the data regarding the total plasma proteins, fibrinogen, and albumin, we assumed that PV may be different in the three MM isotypes, also considering the diverse percentage contribution that each of the protein fractions seemed to exert.

This analysis could underline the importance of blood viscosity in MM, which underwent important changes as the prognostic factors considered varied. In future work, to make the different MM isotypes numerically homogeneous and better deepen the hemorheological evaluation, we will collect a greater study sample, also considering other important information such as that relating to cytogenetic abnormalities, which were currently not evaluated.

## Figures and Tables

**Table 1 biomedicines-11-00964-t001:** Characteristics of the patients.

Parameters	Mean/Percentage
Sex	
Male	46% (88/190)
Female	54% (102/190)
Age at diagnosis	69 ± 10
ISS stage	
Stage I	22% (41/190)
Stage II	26% (49/190)
Stage III	52% (100/190)
Isotype	
IgA k	15% (28/190)
IgA λ	15% (28/190)
IgG k	37% (71/190)
IgG λ	20% (36/190)
Light chain k	6% (13/190)
Light chain λ	7% (14/190)
LDH U/L (normal range: 50 U/L–250 U/L)	193 ± 91
Calcemia mg/dL (normal range:8.6 mg/dL–10.21 mg/dL)	9.57 ± 1.03
Serum creatinine mg/dL (normal range: 0.51 mg/dL–0.95 mg/dL)	1.55 ± 1.58
Monoclonal component g/L	26.01 ± 20.42
Thrombotic risk based on IMWG/NCCN guidelines	
Standard risk	28% (53/190)
High risk	72% (138/190)

**Table 2 biomedicines-11-00964-t002:** Medians, IQRs, and ranges of hemorheological determinants in MM patients.

All MM (*n* = 190)	Median (IQR)	Range
Ht %	31.35 (9.05)	21.00–46.70
Total plasma proteins (g/L)	78.30 (27.50)	46.40–129.6
cWBV 208 s^−1^ (mPa·s)	16.73 (4.25)	10.73–25.35
Fibrinogen (g/L)	3.200 (1.358)	1.090–8.170
Albumin (g/L)	36.70 (10.12)	15.00–48.20
Albumin/fibrinogen ratio	10.88 (5.69)	3.24–28.32

IQR = interquartile range; MM = multiple myeloma; Ht = hematocrit; cWBV = calculated whole-blood viscosity.

**Table 3 biomedicines-11-00964-t003:** Medians (IQRs) of hemorheological determinants in MM patients subdivided according to the isotype of the disease.

	LCMM (*n* = 27)	IgA (*n* = 56)	IgG (*n* = 107)	KWS	*p*
Ht %	31.40 (11.70)	30.35 (8.42)	31.90 (8.80)	0.165	0.4285
Total plasma protein (g/L)	64.20 (6.00)	79.55 (3.47) ***	81.90 (25.50) ***	41.37	<0.0001
cWBV 208 s^−1^ (mPa·s)	14.81 (2.40)	17.71 (4.63) ***	17.35 (3.76) ***	36.18	<0.0001
Fibrinogen (g/L)	3.680 (1.580)	2.870 (1.510) ***	3.200 (1.270) *	15.26	0.0005
Albumin (g/L)	39.90 (5.30)	33.85 (8.37) ***	37.00 (10.20) ***	22.51	<0.0001
Albumin/fibrinogen ratio	11.24 (4.428)	12.40 (5.667)	10.35 (5.401)	2.490	0.2879

IQR = interquartile range; MM = multiple myeloma; LCMM= light chain multiple myeloma; KWS = Kruskal–Wallis statistic; Ht = hematocrit; cWBV = calculated whole-blood viscosity. * *p* < 0.05, *** *p* < 0.001 vs. LCMM (Dunn’s multiple comparison test).

**Table 4 biomedicines-11-00964-t004:** Medians (IQRs) of hemorheological determinants in MM patients subdivided according to ISS.

	ISS Stage I (*n* = 41)	ISS Stage II (*n* = 49)	ISS Stage III (*n* = 100)	KWS	*p*
Ht %	36.70 (9.45)	33.00 (6.90)	29.45 (6.87) ***^,#^	24.63	<0.0001
Total plasma protein (g/L)	73.20 (14.80)	78.30 (24.35)	84.60 (31.40) *	6.516	0.0385
cWBV 208 s^−1^ (mPa·s)	16.55 (3.14)	16.48 (3.60)	17.68 (4.98)	2.591	0.2738
Fibrinogen (g/L)	2.980 (1.185)	3.280 (1.150)	3.180 (1.627)	4.237	0.1202
Albumin (g/L)	40.20 (4.70)	37.00 (9.50) ***	33.55 (9.37) ***^,#^	42.99	<0.0001
Albumin/fibrinogen ratio	12.90 (5.92)	10.35 (4.99) **	9.987 (5.595) ***	20.00	<0.0001

IQR = interquartile range; MM = multiple myeloma; ISS = International staging system; KWS = Kruskal–Wallis statistic; Ht = hematocrit; cWBV = calculated whole-blood viscosity. * *p* < 0.05, ** *p* < 0.01, *** *p* < 0.001 vs. ISS stage I (Dunn’s multiple comparison test). ^#^
*p* < 0.05 vs. ISS stage II (Dunn’s multiple comparison test).

**Table 5 biomedicines-11-00964-t005:** Medians (IQR) of hemorheological determinants in all MM patients subdivided according to the median of albumin.

All MM (n = 190)	Albumin < Median (*n* = 94)	Albumin ≥ Median (*n* = 96)	*p*
Ht %	29.45 (7.80)	32.90 (8.80)	<0.0001
Total plasma proteins (g/L)	83.00 (32.00)	74.70 (18.50)	0.0191
cWBV 208 s^−1^ (mPa·s)	17.49 (5.10)	16.48 (3.11)	0.1375
Fibrinogen (g/L)	3.200 (1.580)	3.190 (1.293)	0.5347
Albumin (g/L)	29.75 (6.85)	39.85 (3.70)	<0.0001
Albumin/Fibrinogen ratio	9.34 (5.217)	12.57 (5.290)	<0.0001

IQR = interquartile range; MM = multiple myeloma; Ht = hematocrit; cWBV = calculated whole-blood viscosity.

**Table 6 biomedicines-11-00964-t006:** Medians (IQR) of hemorheological determinants in all MM patients subdivided according to the median of beta2-MG.

All MM (n = 190)	Beta2-MG < Median (*n* = 94)	Beta2-MG ≥ Median (*n* = 96)	*p*
Ht %	33.95 (9.05)	29.55 (6.78)	<0.0001
Total plasma proteins (g/L)	74.80 (19.17)	82.50 (30.52)	0.0154
cWBV 208 s^−1^ (mPa·s)	16.39 (3.28)	17.29 (4.79)	0.1698
Fibrinogen (g/L)	3.060 (1.028)	3.315 (1.837)	0.0776
Albumin (g/L)	38.80 (8.44)	34.25 (9.05)	<0.0001
Albumin/fibrinogen ratio	12.25 (5.016)	9.52 (5.245)	0.0005

IQR = interquartile range; MM = multiple myeloma; beta2-MG = beta2-microglobulin; Ht = hematocrit; cWBV = calculated whole-blood viscosity.

**Table 7 biomedicines-11-00964-t007:** Medians (IQR) of hemorheological determinants in all MM patients subdivided according to the median of RDW%.

All MM (n = 190)	RDW% < Median (*n* = 92)	RDW% ≥ Median (*n* = 98)	*p*
Ht %	34.90 (9.17)	29.00 (5.80)	<0.0001
Total plasma proteins (g/L)	76.15 (19.48)	78.70 (33.60)	0.0117
cWBV 208 s^−1^ (mPa·s)	16.84 (3.00)	16.60 (5.57)	0.2567
Fibrinogen (g/L)	3.220 (1.262)	3.190 (1.400)	0.7968
Albumin (g/L)	38.60 (6.00)	33.00 (9.50)	<0.0001
Albumin/fibrinogen ratio	11.35 (5.463)	10.06 (5.543)	0.0147

IQR = interquartile range; MM = multiple myeloma; RDW = Red blood cells distribution width; Ht = hematocrit; cWBV = calculated whole-blood viscosity.

**Table 8 biomedicines-11-00964-t008:** Medians (IQR) of hemorheological determinants in all MM patients subdivided according to the median of BMPC%.

All MM (n = 190)	BMPC% < Median (*n* = 92)	BMPC% ≥ Median (*n* = 98)	*p*
Ht %	32.70 (9.50)	30.25 (8.35)	0.0024
Total plasma proteins (g/L)	72.25 (19.50)	81.10 (31.85)	0.1470
cWBV 208 s^−1^ (mPa·s)	16.48 (3.43)	17.22 (5.18)	0.2612
Fibrinogen (g/L)	3.200 (1.275)	3.190 (1.435)	0.4896
Albumin (g/L)	37.35 (9.95)	35.40 (9.70)	0.0949
Albumin/fibrinogen ratio	10.91 (5.734)	10.88 (5.709)	0.3087

IQR = interquartile range; MM = multiple myeloma; BMPC = bone marrow plasma cell; Ht = hematocrit; cWBV = calculated whole-blood viscosity.

## Data Availability

The data presented in this study are available upon request from the corresponding author.

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
