# Peer review of "Calculated Whole Blood Viscosity and Albumin/Fibrinogen Ratio in Patients with a New Diagnosis of Multiple Myeloma: Relationships with Some Prognostic Predictors"

_biomedicines, 2023, doi:10.3390/biomedicines11030964_

Round 1
Reviewer 1 Report (Previous Reviewer 4)
The revised manuscript has changed a lot according to the reviewers’ suggestions. However, a major issue still presents that this study failed to answer the prognostic role of WBV in MM. Though the authors highlight their aim was to evaluate any changes in calculated whole blood viscosity in relation to the changes of prognostic factors, considered at the time of diagnosis (such as ISS). It cannot make up the lack of clinical impact of WBV in current study. Furthermore, the authors should take the doubts of reviewer on p-value selection, it is statistic definition which cannot be changed in relation to your numbers of prognostic factors and sample size.
Author Response
At the moment, the available data do not allow us to conclude about the prognostic role of WBV in MM. In fact, for this specific aim, a larger sample and a patients follow-up long enough for a survival analysis would be needed. However, in our opinion, the current impossibility of attributing a prognostic role to the calculated blood viscosity does not reduce the clinical impact of the paper. In fact, our research represents a preliminary analysis, and, at the same time, a starting point for a subsequent study, certainly prospective, aimed at specifically evaluating the possible prognostic role of WBV in MM setting. With regard to the p-value, we understand that the statistical significance is valid for values of p ≤0.05, but, performing our analysis on a numerically inhomogeneous sample (107 patients with IgG MM, 56 with IgA and 27 with LCMM), and considering different parameters, we have made this doctrinal forcing with the aim to evaluate also the "tendential" statistical significance of the analysed parameters, especially in the evaluation of MM subgroups (IgA, IgG and LCMM).
Reviewer 2 Report (Previous Reviewer 3)
This is a resubmission of a manuscript after major revision. The topic of plasma viscosity as determined by serum albumin and fibrinogen levels is an interesting and import topic in myeloma care. I maintain my attitude that blood viscosity in active myeloma is determined by serum paraproteins present. On the other hand, I do accept the conclusions of the authors with one important caveat: their measurements have relevance for myeloma in remission with low M-protein levels. I maintain that some actual confirmatory viscosity measurements would greatly add to the value of the paper.
Author Response
We absolutely agree on the significant influence of M-protein on blood viscosity, but in this analysis, in which we also studied patients with lower paraprotein values at the diagnosis phase and patients with LCMM, we considered other important factors with a recognized impact on blood viscosity. Regarding the direct measurements, we well understand their clinical relevance, but, faced with the difficulty of carrying them out (availability and maintenance of the necessary instrumentation as well as technicians expert in their use) and also encouraged by the results of our previous research (doi: 10.3233/CH-211198), we believe that the possibility of using calculated parameters can be a great advantage in clinical evaluation of some aspects of the hemorheological of our patients.
Reviewer 3 Report (Previous Reviewer 1)
Corrections have been made. Manuscript should be accepted for publication now.
Author Response
Thank you for your agreement regarding the publication of our paper.
Reviewer 4 Report (Previous Reviewer 2)
This revised manuscript was described well, and so I considered that this revised manuscript was suitable to publish for "Biomedicines".
Author Response
Thank you for your agreement regarding the publication of our paper.
Round 2
Reviewer 1 Report (Previous Reviewer 4)
Accept in present form.
This manuscript is a resubmission of an earlier submission. The following is a list of the peer review reports and author responses from that submission.
Round 1
Reviewer 1 Report
Melania Carlisi and colleagues present a quality and well-written experimental manuscript focused on calculated whole blood viscosity and albumin/fibrinogen ratio in patients with new diagnosis of multiple myeloma with regards to relationships with some prognostic predictors.
Authors aimed to evaluate possible relationships with some prognostic predictors, as ISS, albumin levels, beta2-microglobulin, red cell distribution width and bone marrow plasma cell infiltration.
For that authors examined a cohort of 190 patients, and the calculated blood viscosity using the de Simone formula, and the albumin/fibrinogen ratio as a surrogate of erythrocyte aggregation, and then they related these parameters to prognostic factors, using respectively the Kruskal-Wallis test and the Mann-Withney test.
Authors found that the evaluated hemorheological pattern differ in the three isotype of multiple myeloma and the whole blood viscosity is higher in IgA and IgG isotypes respect to the light chain multiple myeloma. Moreover, they observed that, as the ISS stage progresses, the albumin/fibrinogen ratio are reduced, and the same hemorheological trend is traced in subgroups with lower albumin levels, higher beta2-microglobulin and RDW values, and in the presence of a greater bone marrow plasma cell infiltrate.
Finally, authors conclude that their analysis might remark the role of blood viscosity in MM, guarantying a more complete and efficient risk assessment that also includes the hemorheological pattern.
Overall, the manuscript is valuable for the scientific community and should be accepted for publication after edits are made.
===========================
Other comments:
1) Please check for typos throughout the manuscript.
2) With regards to novel treatments of multiple myeloma – authors are kindly encouraged to cite the following review that describes cell immunotherapy based approach for MM therapy. DOI: 10.3390/cancers14041078
Reviewer 2 Report
Biomedicines-2211527
Calculated whole blood viscosity and albumin/fibrinogen ratio in patients with new diagnosis of multiple myeloma: relationships with some prognostic predictors.
The original article “Calculated whole blood viscosity and albumin/fibrinogen ratio in patients with new diagnosis of multiple myeloma: relationships with some prognostic predictors. " (biomedicines-2211527) by Carlisi M, et al. demonstrated that WBV, calculated using Simone formula, was different among the M protein isotype in myeloma patients. In addition, albumin and fibrinogen level were analyzed for several subgroup, such as ISS stage. This article revealed the relationship between WBV and several biomarkers, which was interesting. However, the clinical significance of WBV for survival time, treatment response, adverse events, and myeloma related symptoms were not investigated. Therefore, the clinical impact of WBV was low, unfortunately. The author should analyze more in detail for the purpose to improving this original article as below.
1. In this article, WBV in the IgA and IgG type myeloma patients was higher than those in the LC myeloma patients. WBV was generally associated with burden of serum M protein, so that these results were little clinical impact. I considered that the author should investigate the clinical significance of WBV for survival time, treatment response, adverse events (especially thrombosis), and myeloma related symptoms for the purpose to improve understanding clinical significance of WBV.
2. After treatment, was WBV value changed? I considered that treatment response was association with increasing Hct and decreasing TP, so that WBV reduced. The author could analyze the clinical impact of WBV change during treatment as well.
3. Why was the cut-off of P-value 0.1?
Reviewer 3 Report
This manuscript attempts to shed light on the whole-blood viscosity problem in multiple myeloma. Unfortunately, it is retrospective work with no actual viscosity measurements. As viscosity may heavily depend not only on the type of M-protein but the amount thereof, such a calculation - may it be very extensive - have a limited role.
Reviewer 4 Report
The manuscript entitled “Calculated whole blood viscosity and albumin/fibrinogen ratio in patients with new diagnosis of multiple myeloma: relationships with some prognostic predictors” has found that the calculated blood viscosity and the albumin/fibrinogen ratio can be used as prognostic factors. The topic is of interest, however, the author fails to present their data and manuscript clearly to show the correlation of these two parameters and the prognosis. Especially, there are some flaws in the manuscript presentation.
1. Line 54, 57,60 not proper punctuation or mixture of beta2 MG or β2MG
2. ISS have standard stage as I, II, III instead of score=1,2,3, et al
3. It’s hard to conclude these two parameters were prognostic factors without survival analysis
4. In methods, its not professional to show the line markers in line 117 to 125. ï‚§
5. Tables in manuscript are not proper presented, the format is not consistent.
Overall, the results are not sufficient to support the conclusion. The study still needs extensive manuscript editing and data support.